# A Mild Method for Encapsulation of Citral in Monodispersed Alginate Microcapsules

**DOI:** 10.3390/polym14061165

**Published:** 2022-03-15

**Authors:** Wen-Long Ma, Chuan-Lin Mou, Shi-Hao Chen, Ya-Dong Li, Hong-Bo Deng

**Affiliations:** College of Chemistry and Chemical Engineering, Southwest Petroleum University, Chengdu 610500, China; long755653@163.com (W.-L.M.); c_shihao0807@163.com (S.-H.C.); edwardools@163.com (Y.-D.L.)

**Keywords:** alginate microcapsules, microfluidic, GDL, citral, mild gelling method

## Abstract

Citral is a typical UV-irritation and acid-sensitive active and here we develop a mild method for the encapsulation of citral in calcium alginate microcapsules, in which UV irritation or acetic acid is avoided. Monodispersed oil-in-water-in-oil (O/W/O) emulsions are generated in a capillary microfluidic device as precursors. The middle aqueous phase of O/W/O emulsions contains sodium alginate, calcium-ethylenediaminetetraacetic acid (EDTA-Ca) complex as the calcium source, and D-(+)-Gluconic acid δ-lactone (GDL) as the acidifier. Hydrolysis of GDL will decrease the pH value of the middle aqueous solution, which will trigger the calcium ions released from the EDTA-Ca complex to cross-link with alginate molecules. After the gelling process, the O/W/O emulsions will convert to alginate microcapsules with a uniform structure and monodispersed size. The preparation conditions for alginate microcapsules are optimized, including the constituent concentration in the middle aqueous phase of O/W/O emulsions and the mixing manner of GDL with the alginate-contained aqueous solution. Citral-containing alginate microcapsules are successfully prepared by this mild method and the sustained-release characteristic of citral from alginate microcapsules is analyzed. Furthermore, a typical application of citral-containing alginate microcapsules to delay the oxidation of oil is also demonstrated. The mild gelling method provides us a chance to encapsulate sensitive hydrophobic actives with alginate, which takes many potential applications in pharmaceutical, food, and cosmetic areas.

## 1. Introduction

Microencapsulation of hydrophobic substances in biopolymers is widely applied in many areas, such as energy storage, food, cosmetics, and the pharmaceutical industry, enabling isolation, protection, and controlled release [1,2,3,4,5]. Citral is a typical hydrophobic chemical with a special aroma, which is usually used as an additive in the food industry and the daily chemical industry because of its antioxidant and antibacterial properties [6,7,8,9,10]. However, citral is susceptible to deterioration by a series of cyclization and oxidation reactions when stimulated by some environmental factors, such as acid, UV light, and high temperature [11,12,13]. Therefore, microencapsulation technology has been used to increase the chemical stability of citral and realize controlled release. The citral-containing microcapsules take many applications in food-spoilage inhibition [14,15] and air purification [16] etc. Many methods have been applied to citral encapsulation, such as oil-in-water and multilayer emulsions, molecular complexes, and spray-drying [17,18,19,20]. Among those technologies, spray-drying is the most commonly used method. However, the activity of citral would be affected by the high temperature during the spray-drying process and the sizes of resulting particles are widely distributed, making the release process of citral from the microcapsules unpredictable.

Compared with spray-drying technology, the microfluidic-based encapsulation method shows high controllability in tailoring the structure and size of the resulting microcapsules [21,22,23]. In combination with microfluidic technology, a variety of monodispersed microcapsules with biomaterials shells are produced to encapsulate hydrophobic substances. Among those biomaterials, alginate is a commonly used and popular shell material because of its convenient sources, nontoxicity, and excellent biocompatibility [24,25,26]. Alginate aqueous solution could be gelled by complexing with multivalent cautions such as Ca^2+^, Ba^2+^, or Fe^3+^, in which Ca^2+^ is more preferable because of low toxicity and cost [27,28,29,30]. Alginate microcapsules with oil cores can be prepared by on-line gelation in microfluidic devices or off-line gelation of oil-in-water-in-oil (O/W/O) double emulsions. Eqbal et al. have prepared alginate microcapsules, each with a tail by on-line merging of alginate-contained O/W/O double emulsions with calcium chloride-contained water-in-oil (W/O) emulsions [31]. Huang et al. have prepared alginate microcapsules with different shapes by using CaCO_3_ nanoparticles-contained O/W/O emulsions as templates and acetic acid as a trigger [32]. The morphologies of the alginate microcapsules from on-line gelation method are commonly hard to control accurately because the fast gelation process has occurred in a flowing state with a dynamic interface [33,34]. Alginate microcapsules with oil cores produced by the off-line gelation method have better sphericity than the ones from the on-line gelation method. Different strategies for off-line gelling O/W/O double emulsions to produce alginate microcapsules have been developed. Liu et al. have prepared alginate microcapsules with oil cores by using a photoacid generator to trigger the Ca^2+^ ions released from the CaCO_3_ nanoparticles with UV irritation [35]. However, the UV irritation will initiate the photoreaction of citral [11]. In our previous study, calcium ethylenediaminetetraacetic acid (EDTA-Ca) complex is used as a water-soluble calcium source and acetic acid as a trigger to prepare alginate microcapsules with oil cores [36]. A water-soluble EDTA-Ca calcium source makes the inner structure of the alginate shells more homogeneous than the ones using CaCO_3_ nanoparticles as a calcium source. However, the acetic acid is soluble in both water and oil and it will diffuse across the alginate shell into the inner oil cores. The dissolved acetic acid in the oil core would induce the acid-catalyzed cyclization of citral. Therefore, a new mild strategy of preparing alginate microcapsules for citral, or a similar sensitive hydrophobic substance, encapsulation is highly desired.

In this paper, we report a method for preparing monodispersed alginate microcapsules with citral under mild conditions, without high temperature, UV irritation, or acetic acid. A two-staged microfluidic device is used to generate monodispersed O/W/O double emulsions (Figure 1a), in whose middle aqueous phase contains sodium alginate, EDTA-Ca as calcium source, and D-(+)-Gluconic acid δ-lactone (GDL) as acidifier (Figure 1b). The hydrolysis of GDL in an aqueous solution will decrease the pH and trigger the calcium ions released from EDTA-Ca (Figure 1c). After the alginate molecules are cross-linked with the liberated calcium ions, the O/W/O emulsions will convert to alginate microcapsules (Figure 1d). The preparation conditions for alginate microcapsules are optimized and citral-containing alginate microcapsules are successfully prepared by this mild method. The sustained-release characteristic of citral from alginate microcapsules is analyzed. Furthermore, a typical application of citral-containing alginate microcapsules to delay the oxidation of oil is also demonstrated.

## 2. Materials and Methods

### 2.1. Materials

Sodium alginate (from brown algae, low viscosity) and Pluronic F-127 (F-127) were obtained from Sigma-Aldrich (Saint Louis, Missouri, America). D-(+)-Gluconic acid δ-lactone (GDL) and ethylenediaminetetraacetic acid disodium salt (disodium–EDTA) were purchased from Aladdin (Shanghai, China). Citral (97%) was gained from Shanghai Macklin Biochemical Co., Ltd. (Shanghai, China). Calcium chloride anhydrous, sodium hydroxide, Sudan III (bioreagent), acetic acid, isooctane, potassium iodide, sodium thiosulfate, starch, and 2, 2, 4-trimethylpentane were acquired from Kelong Chemical Reagent Factory (Chengdu, China). Polyglycerol polyricinoleate (PGPR 90) was purchased from Danisco (Kunshan, China). Soybean oil (SO) was provided by Sichuan Jiali Grain and Oil Co., Ltd. (Chengdu, China). Benzyl benzoate (BB) was purchased from Sinopharm Chemical Regent Co., Ltd. (Shanghai, China). Cylindrical capillary tubes and square capillary were obtained from VitroCom (New York, NY, American). Deionized (DI) water from a water purification machine (Ulupure, Sichuan ULUPURE Ultrapure Technology Co.,Ltd., Chengdu, China) was employed in the experiments. All the reagents were of analytical grade and used as received unless otherwise stated.

### 2.2. Microfluidic Devices

A two-staged 3D glass capillary microfluidic device was assembled to generate O/W/O double emulsions as show in Figure 1a [23,37,38]. The outer diameters of all the cylindrical capillary tubes are 0.96 mm and the square capillary tubes have an inner dimension of 1 mm × 1 mm. The inner diameters of the injection tube, the transition tube and collection tube are 0.55 mm, 0.22 mm, and 0.4 mm, respectively. A micropuller (PN-30, Narishige, Tokyo, Japan) and a microforge (MF-830, Narishige, Tokyo, Japan) are used to tailor the outlets of the injection tube and transition tube. The inner diameters of the outlets of the injection tube and transition tube are 0.06 mm and 0.14 mm, respectively. At each end of the square capillary tube, epoxy resin glue is used to fix and seal the tubes.

A Y-shaped micro-mixer was also designed for on-line mixing of GDL aqueous solution with sodium alginate-contained aqueous solution as a middle fluid, as shown in Appendix A. Stainless steel cylindrical tubes with two different sizes and polyethene tubes were used to construct the micro-mixer. The inner and outer diameters of the small stainless steel cylindrical tube are 0.5 and 0.7 mm, respectively, while those of the bigger one are 1 mm and 1.26 mm. The small stainless steel cylindrical tube is concentrically inserted into a bigger one with a polyethene pipe connected to the end and the pipes are fixed and sealed with epoxy resin glue. When the GDL aqueous solution and alginate-contained aqueous solution are injected into the inlets of the Y-shaped micro-mixer separately, the two streams will be mixed before flowing into the microfluidic device.

### 2.3. The Hydrolysis Processes of GDL

GDL is used as an acidifier to trigger the release of Ca^2+^ ions from EDTA-Ca. The released Ca^2+^ ions will cross-link with alginate molecules rapidly [33]. Thus, the gelling speed of alginate aqueous solution is mainly controlled by the hydrolysis process of GDL.

The hydrolysis process of GDL in DI water is investigated. GDL (2%, *w*/*v*) is dispersed into DI water at room temperature and the pH variation of the solution is monitored with a pH meter (Leici PHS-3C, Chengdu century Fangzhou Technology Co.,Ltd., Chengdu, China).

At the same time, the hydrolysis processes of GDL in alginate-contained solution is also investigated. Calcium chloride (0.1 M) and disodium–EDTA (0.1 M) are dissolved in DI water and then the pH value of the mixture is adjusted to about 7.0 with sodium hydroxide (2 M). EDTA^2−^ will complex the calcium ions to form EDTA-Ca in the mixture [39]. The equivalent concentration of EDTA-Ca in the aqueous solution is 0.1 M. Sodium alginate (2%, *w*/*v*) are added into the EDTA-Ca contained aqueous solution and mixed with a magnetic mixer until sodium alginate is totally dissolved. GDL (2%, *w*/*v*) is added into the mixture and the pH value change the solution is also recorded by a pH meter.

### 2.4. Influence of Constituent Concentration on the Gelling Time of Alginate Aqueous Solution

Aqueous solution with alginate (2%, *w*/*v*), EDTA-Ca (0.1 M) and GDL 2% (*w*/*v*) is prepared with the same method as Section 2.3. The hydrolysis of GDL will make the calcium ions released from EDTA-Ca complex and trigger the gelling process. The GDL-contained aqueous solution (2 mL) is introduced into a baker (10 mL). The gelling state of the aqueous solution is analyzed by inclining the beaker at different time to observe the flow behaviors.

Influence of constituent concentration on the gelling time of the aqueous solution is carried out by varying the concentration of GDL, sodium alginate, and EDTA-Ca individually. The concentration of GDL in the mixture is adjusted from 0.5% (*w*/*v*) to 5% (*w*/*v*), while the concentration of other components remains unchanged. The gelling time of aqueous solution with varied GDL content is recorded. The influences of the sodium alginate and EDTA-Ca concentrations on the gelling time of the alginate aqueous solution are also investigated with the same method. The concentration of sodium alginate in the mixture is varied from 1% (*w*/*v*) to 3% (*w*/*v*) and the gelling time is compared. Similarly, the concentration of EDTA-Ca is changed by varying that of the calcium chloride and disodium–EDTA synchronously. The concentrations of calcium chloride and disodium–EDTA are varied from 0.01 M to 0.2 M and the gelling time is also compared.

### 2.5. Influence of Constituent Concentration on the Formation of Alginate Microcapsules

The inner fluid is a mixture of soybean oil and benzyl benzoate with a volume ratio of VSO: VBB = 1:1 containing 2% (*w*/*v*) PGPR 90. Sudan III is also added in the inner fluid with a concentration of 0.1% (*w*/*v*) as an oil-soluble dye. The outer fluid and the collection fluid are soybean oil containing 5% (*w*/*v*) PGPR 90. Calcium chloride (0.1 M) and disodium–EDTA (0.1 M) are dissolved in DI water and then the pH value of the mixture is adjusted to about 7.0 with sodium hydroxide (2 M). The equivalent concentration of EDTA-Ca in the aqueous solution is 0.1 M. Sodium alginate (2%, *w*/*v*) and F-127 (0.5%, *w*/*v*) are added into the aqueous solution with EDTA-Ca and mixed with a magnetic mixer until sodium alginate is totally dissolved. GDL (2%, *w*/*v*) is added into the mixture and used as middle fluid. Unless otherwise stated, these fluids are used in the following microfluidic experiment to prepare of O/W/O emulsions.

The influences of sodium alginate, EDTA-Ca and GDL on the formation of alginate microcapsules are investigated by varying the concentration independently. The concentration of GDL in the middle fluid is varied from 1% (*w*/*v*) to 3% (*w*/*v*) or that of the sodium alginate is varied from 1% (*w*/*v*) to 3% (*w*/*v*). The concentration of EDTA-Ca in the middle fluid is changed by varying that of the calcium chloride and disodium–EDTA synchronously. Calcium chloride and disodium–EDTA are also varied from 0.1 M to 0.05 M. The formation of O/W/O in microfluidic device and the gelling process are analyzed with an optical microscope.

### 2.6. Preparation of Alginates Microcapsules by On-Line Mixing

The Y-shaped micro-mixer could on-line mix GDL solution with sodium alginate-contained solution to produce middle fluid for O/W/O emulsions preparation, as shown in Appendix A. Calcium chloride (0.2 M) and disodium–EDTA (0.2 M) are dissolved in DI water and then the pH value of the mixture is adjusted to about 7.0 with sodium hydroxide (2 M). Sodium alginate (4%, *w*/*v*) and F-127 (1%, *w*/*v*) are added into the aqueous solution and mixed with a magnetic mixer until sodium alginate is totally dissolved. The sodium alginate-contained aqueous solution is injected into one inlet of the Y-shaped micro-mixer with a syringe pump at 4 μL/min, while the GDL (4%, *w*/*v*) aqueous solution is pumped into another inlet of the micro-mixer with the same flow rate. The alginate-contained solution and GDL solution will be mixed in Y-shaped micro-mixer before flow into the microfluidic device. The flow rates of the inner fluid and outer fluid are 2.6 μL/min and 30 μL/min, respectively. The O/W/O emulsions from microfluidic are collected in a collection fluid.

### 2.7. Preparation of Alginate Microcapsules with Citral

Soybean oil and benzyl benzoate are mixed with a volume ratio of V_SO_:V_BB_ = 1:1, and then PGPR 90 (2%, *w*/*v*), citral (1%, *w*/*v*) are added. The citral-containing solution is mixed under dark conditions and used as an inner fluid. The middle fluid and the outer fluid is the same as in Section 2.5. The flow rates of the inner fluid, middle fluid, and outer fluid are 2 μL/min, 4 μL/min, and 24 μL/min, respectively. The generated O/W/O emulsions are gathered in a collection oil solution. After 3 h of reaction in the collection oil solution, the formed alginate microcapsules are separated from the collection solution by a calcium chloride solution (0.1%, *w*/*v*) and DI water. The water-soluble components in alginate shell will be removed in the repeated washing process. The alginate microcapsules are enhanced by being further cross-linked in calcium chloride (1%, *w*/*v*) solution. The morphologies and size distributions of O/W/O emulsions and alginate microcapsules are analyzed with an optical microscope.

### 2.8. The Sustained Release of Citral from Alginate Microcapsules

The release process of citral from alginate microcapsules in ethanol solution is investigated. Citral shows poor solubility in water, which makes the release of citral from the microcapsules very slow in an aqueous solution. At the same time, the released citral is easy to volatilize. The accuracy of the experimental data will be greatly affected by citral volatilization because of the long-time release process. Citral is soluble in organic solvents, such as ethanol. Therefore, the sustained-release of citral-containing microcapsules is commonly carried out in an ethanol solution [40]. Citral-containing alginate microcapsules (0.5 mL) are prepared as described in Section 2.7 and added into a conical flask with 200 mL of 40% (*v/v*) ethanol solution. The conical flask is vibrated in a water bath oscillator with a rotational speed of 100 r/min. The supernatant (2 mL) is taken out at specific time intervals to measure the concentration of citral and, at the same time, 2 mL of 40% (*v/v*) ethanol solution is added into the beaker with the citral-containing microcapsules to keep the total volume unchanged. The concentration of citral in the ethanol solution is tested with UV/Vis spectroscopy.

The total amount of the citral released from the microcapsule in t (Mt) and the cumulative release percentage (R) are calculated based on the UV/Vis spectroscopy test results [40,41].
(1)$$M_{i}=VC_{i}+\sum C_{i-1}V_{s}$$
(2)R=MtM×100%
where $C_{i}$ and $C_{i-1}$ are the concentration of citral at the *i*, *i*−1 sampling point (μL/mL), *V*: initial volume of the solution (200 mL), $V_{s}$: the sampling volume for UV/Vis spectroscopy test (2 mL), *M*: total citral released from the microcapsules in 3 h.

The zero-order release equation, first-order release equation, Higuchi plane diffusion equation, and Retger-peppas equation are utilized to fit the release process [42,43].

### 2.9. Antioxidant Properties of Oil with Citral-Containing Microcapsules

In order to investigate the influence of antioxidation activity of citral to oil, the peroxide values (POV) of soybean oil with citral-containing alginate microcapsules, with dissolved citral, and without citral are compared.

The method of preparing citral-containing alginate microcapsules is the same as in Section 2.7. The O/W/O emulsions are gathered in collection fluid for 30 min, and the total volume of the mixture is increased to about 40 mL by adding the collection fluid. When the O/W/O emulsions convert to alginate microcapsules, the citral will release into the continuous oil phase in a sustained manner. According to the flow rate of the inner fluid and the duration of emulsion collection, the volume of the citral-containing inner fluid in alginate microcapsules is about 60 μL in total. At the same time, 40 mL of collection fluid with 60 μL inner fluid and 40 mL of collection fluid without any citral are used as control groups.

Potassium iodide will react with peroxides in oil to generate precipitated iodine under acidic conditions. The precipitated iodine is titrated by sodium thiosulfate, and the peroxide value of the oil is calculated based on the amount of sodium thiosulfate [44]. The tested oil (1 mL) is added to the mixture of isooctane (20 mL) and glacial acetic acid (30 mL), and then saturated potassium iodide solution (1 mL) is added. After 1 min, distilled DI water (50 mL) is added and then sodium thiosulfate solution (0.01 M) is continuously dripped into the mixture until the solution fades to pale yellow. Starch indicator (1 mL) is added into the solution to guide the titration end point of sodium thiosulfate solution. The sodium thiosulfate solution is dripped into the mixture drop by drop until the blue color disappears. The peroxide value of the oil samples is measured every 24 h.

### 2.10. Characterization

The morphologies of the O/W/O emulsions and alginate microcapsules are observed with an optical microscope (IX31, Olympus, Tokyo, Japan) with a CCD camera. The sizes and size distributions are determined using commercial size-analysis software (JIFEI, Nanjin Yifei Science and Technology Ltd., Nanjing, China). The monodispersity of emulsion droplets or microcapsules is evaluated based on an index called the coefficient of variation (*CV*), which is defined as the ratio of the standard deviation of size distribution to its arithmetic mean ($\bar{D}$).
(3)$$\bar{D}=\overset{N}{\underset{1}{\sum }}D_{i}/N$$
(4)CV=∑i=1NDi−D¯2N−112D¯
where $D_{i}$ is the diameter of the emulsion droplets or microcapsules and *N* is the total number of the emulsion droplets or microcapsules counted. In this study, more than 100 emulsion droplets or microcapsules are counted for obtaining each average diameter and *CV* value. Commonly, when the *CV* value is no more than 5%, the emulsions or particles are supposed to be monodispersed.

## 3. Results and Discussion

### 3.1. The pH Change Kinetics of GDL-Contained Solution

GDL is widely used as a delayed-action acidifier during food processing and could be partially hydrolyzed to generate gluconic acid, which is a kind of weak acid [45,46]. The pH of GDL (2%, *w*/*v*) aqueous solution at room temperature is 3.4, which will decrease to 2.8 gradually within about 40 min, as shown in Figure 2a. However, the pH variation becomes quite slow afterwards and the pH value is about 2.6 at 150 min. The Ca^2+^ ions will be released from the EDTA-Ca complex and cross-link with alginate molecules when the pH value is lower than 4.8 [36]. Premature releasing of the Ca^2+^ ions from the EDTA-Ca complex to cross-link with alginate molecules will disable the preparation of O/W/O emulsions by a microfluidic method. Therefore, the pH variation of the aqueous solution with sodium alginate-contained (2%, *w*/*v*), EDTA-Ca (0.10 M), and GDL (2%, *w*/*v*) is also investigated, as shown in Figure 2b. The initial pH value of the mixture is about 5.7, which is evidently higher than that of the GDL (2%, *w*/*v*) aqueous solution. After 40 min, the pH value of EDTA-Ca-contained solution decreases to 4.8 and the Ca^2+^ ions would start to release from the EDTA-Ca complex. EDTA is a kind of weak acid; therefore, the EDTA^2−^ will partially bind with hydrogen ions from the hydrolysates of GDL. Thus, the initial pH value of EDTA-Ca-contained GDL aqueous solution is higher than that of GDL (2%, *w*/*v*) aqueous solution. The delayed release of Ca^2+^ ions from EDTA-Ca provides us a chance to prepare O/W/O emulsion precursors for alginate microcapsules.

### 3.2. The Influence of Constituent Concentration on the Gel Time of Alginate Hydrogel

The gelling time of alginate-contained aqueous solution is influenced by the concentration of GDL, sodium alginate, and EDTA-Ca, as shown in Figure 3.

When sodium alginate and EDTA-Ca are fixed at 2% (*w*/*v*) and 0.1 M, respectively, the gelling time decreases along with the increasing of GDL concentration (Figure 3a and Appendix A). When the GDL concentration is 0.5% (*w*/*v*), the alginate-contained aqueous solution will not gel even after 48 h. The initial gelling time of an alginate aqueous solution is about 180 min when the GDL is 1% (*w*/*v*). With the increase of GDL concentration to 5% (*w*/*v*), the initial gelling times are shortened to 15 min. However, too-fast gelation of the alginate aqueous is unfavorable for preparation of O/W/O in a microfluidic device. Therefore, the concentration of GDL should be smaller than 3% (*w*/*v*) to avoid the unexpected blocking of the microfluidic device. Similarly, when the concentrations of GDL (2%, *w*/*v*) and EDTA-Ca (0.10 M) are fixed, the gelling times will decrease from 150 min to 30 min while the alginates sodium increases from 1% (*w*/*v*) to 3% (*w*/*v*), as shown in Figure 3b.

However, the gelling time will extend from 30 min to 115 min when the concentration of EDTA-Ca rises from 0.01 M to 0.2 M while the alginates sodium and GDL are the same of 2% (*w*/*v*). The increasing of EDTA-Ca will delay the release of Ca^2+^ from the complex. The solubility of the EDTA-Ca in water largely depends on the concentration and the pH value. EDTA-Na_2_ (0.1 M) and CaCl_2_ (0.1 M) could totally dissolved in DI water before or after the pH value is adjusted to 7.0, and the equivalent concentration of EDTA-Ca is 0.1 M (as shown in Appendix A). However, when the concentration of EDTA-Na_2_ and CaCl_2_ are raised to 0.2 M synchronously, an insoluble milky white suspended subtract will be observed (as shown in Appendix A). The suspended subtract will dissolve as the pH of the aqueous solution is adjusted to about 7.0 (Appendix A). Further raising the concentration of EDTA-Na_2_ and of CaCl_2_ to 0.3 M, insoluble precipitation will appear (Appendix A).

### 3.3. Influence of Alginate Sodium, GDL, and EDTA-Ca on the Formation of O/W/O Emulsions and Alginate Microcapsules

Different from the producing of alginate hydrogel, the preparation of alginate microcapsules will be subject to the formation of O/W/O emulsions in the microfluidic device and the stability of the transition process from O/W/O emulsion precursors to microcapsules.

O/W/O emulsions with different content of alginate sodium but similar sizes are prepared by the regulation of flow rates in microfluid (as shown in Appendix A). In the middle aqueous layer of O/W/O emulsions, the concentration of alginate sodium ranges from 1% (*w*/*v*) to 3% (*w*/*v*), while the GDL and EDTA-Ca are fixed at 2% (*w*/*v*) and 0.1 M respectively. All the emulsions can convert to alginate microcapsules successfully (Appendix A). However, the viscosity of the middle fluids for O/W/O emulsions preparation is mainly affected by the concentration of alginate sodium. The viscosity of the aqueous solution will rise with the concentration of sodium alginate, as shown in Figure 4. When the middle layer of O/W/O double emulsions takes higher viscosity, the corresponding O/W/O emulsions will be more stable in the gelling process [23]. However, middle fluids with excessive alginate sodium concentration, such as 3% (*w*/*v*) or more, is difficult to shear dispersing in microfluidic device due to the high viscosity [47]. The middle fluid with 2% (*w*/*v*) alginate sodium takes good comprehensive performance both for microfluidic operation and emulsion stability.

O/W/O emulsions with 2% (*w*/*v*) alginate sodium, 0.10 M EDTA-Ca and varied GDL content in the middle aqueous layers are also prepared from microfluidic device, as shown in Figure 5(a1,b1,c1). The gelling processes of the O/W/O emulsions are affected by the concentration of GDL evidently. When the GDL is 1% (*w*/*v*) in the middle aqueous phase, the emulsions will became quit unstable in the gelling process. Almost all the inner oil droplets will coalescence with the continuous oil phase and the O/W/O emulsions are evolved to W/O emulsions (Figure 5c). The coalescence of inner droplets with the continuous phase is a typical form of demulsification during emulsion polymerization process [23]. In Figure 5(c2), colored oil could be observed around the demulsified emulsions, which was due to coalescence of the red inner oil droplets with the continuous oil phase. Furthermore, the outer diameters of W/O emulsions based on the size of the O/W/O emulsions and the inner oil droplets by volume is calculated. The outer diameter of O/W/O emulsions will decrease 11.9% after demulsification to W/O emulsions. The outer diameters of O/W/O emulsions and the corresponding W/O emulsions are measured. The result shows that the average outer diameter of O/W/O emulsions and the corresponding W/O emulsions are 311.7 μm and 274.4 μm, respectively. The diameter of the W/O emulsions is 12% less than that of the O/W/O emulsions, which is basically consistent with the results theoretical calculation. The stability of the emulsions are improved as the GDL increases to 1.5% (*w*/*v*) and 2% (*w*/*v*), as shown in Figure 5a,b. Conversion ratio from O/W/O emulsions to alginate microcapsules are about 55.2% and 98.1% when the GDL concentrations are 1.5% (*w*/*v*) and 2% (*w*/*v*), respectively.

As the GDL is premixed with alginate sodium and the EDTA-Ca contained aqueous solution as the middle fluid, the hydrolysis of GDL and gelling reaction will start during the formation of O/W/O emulsions in the microfluidic device. After about 50 min of premixing of GDL, the formation of O/W/O emulsions in the microfluidic device become unstable and, at the same time, the microfluidic device also takes the risk of blocking of hydrogel. With the further increase of the content of GDL to 3% (*w*/*v*), alginate microcapsules could also be produced (as shown in Appendix A). However, the duration of O/W/O emulsions formed stably in the microfluidic device will decrease.

The concentration of EDTA-Ca will also affect the gelling process. In theory, the mechanical strength of the resulting alginate microcapsules could be strengthened by increasing the calcium ions. Thus, a higher concentration of EDTA-Ca is preferred. However, the content of EDTA-Ca is limited by its solubility. The EDTA-Ca concentration is limited to 0.1 M to prevent insoluble matter from producing defects in the alginate microcapsules.

### 3.4. Morphology and Size Analysis of O/W/O Emulsions and Alginate Microcapsules

Based on the results in Section 3.3, with an optimized recipe (Appendix A), O/W/O double emulsion precursors are produced from the microfluidic device, as shown in Figure 6a. The corresponding size distribution of O/W/O emulsions and the inner oil droplets is shown in Figure 6b. The average internal and external diameters of the double emulsions are 156 μm (CV = 3.88%) and 267 μm (CV = 2.96%), respectively, which means the O/W/O double emulsions are monodispersed. The GDL in the middle aqueous phase of O/W/O emulsions will hydrolyze slowly to trigger the calcium ions released from EDTA-Ca and cross-linked with alginate molecules. After about 3 h, the O/W/O double emulsions will transit to alginate microcapsules in a stable manner in the collection solution. The alginate microcapsules are separated into DI water, and then being further cross-linked with 1% (*w*/*v*) calcium chloride solution to limit the excessive swelling of alginate microcapsules (Appendix A), which may cause accidental leakage. The outer and inner size of the microcapusles, as well as the thickness of the shell, will slightly decrease after further crosslinking. The average inner and outer diameters of the further cross-linked alginate microcapsules are 163 μm (CV = 3.78%) and 230 μm (CV = 3.11%), respectively.

### 3.5. Preparation of Microcapsules Based on Micro-Mixer

In the process of O/W/O emulsions preparation, hydrogel will slowly appear in the syringe with middle fluid after about 60 min of premixing. The unexpected gelling is due to the premixing of GDL with alginate-contained aqueous solution, which will limit the continuous and large-scale preparation of alginate microcapsules with this method. The unexpected gelling can be eliminated by improving the mixing manner of GDL with alginate-contained aqueous solution. The GDL solution and alginate-contained aqueous solution could be on-line mixed with the Y-shaped micro-mixer.

The GDL aqueous solution and alginate-contained aqueous solution are infused into the inlets of Y-shaped micro-mixer and the two stream will be mixed before being infused into the microfluidic device. The GDL solution could be updated every half an hour and unexpected gelling will not appear anymore. The formed O/W/O emulsions are stable and uniform as the ones by the pre-mixing manner (Figure 7a). The diameters of O/W/O emulsions and the inner oil droplets distributed in a relatively small range (Figure 7b). The average diameters of the O/W/O emulsions and the inner oil droplets are 267 μm (CV = 2.4%) and 147 μm (CV = 2.8%), respectively. The resulting microcapsules are also uniform and integrated in structure (Figure 7c). The size distributions of the inner and outer diameters of the alginate microcapsules are shown in Figure 7d. The average inner and outer diameters of the alginate microcapsules are 165 μm (CV = 3.4%) and 243 μm (CV = 2.9%), respectively. The O/W/O emulsions and alginate microcapsules from the on-line mixing method are as monodispersed as those from the pre-mixing manner.

The results indicated that the shift of the mixing manner from pre-mixing to on-line mixing will eliminate the unexpected gelling but have little influence on the resulting O/W/O emulsions and alginate microcapsules. Furthermore, in the process of industrialized production, the GDL aqueous solution can be continuously prepared and injected into a Y-shaped micro-mixer, enabling the continuous and large-scale preparation of alginate microcapsules with hydrophobic substances.

### 3.6. Sustained-Release from Citral-Containing Alginate Microcapsules

Monodispersed O/W/O emulsions with citral-containing inner oil cores are prepared with microfluidic device (Appendix A). After the gelling process, O/W/O emulsions converted to citral-containing alginate microcapsules. The citral-containing alginate microcapsules are separated into DI water and further cross-linked with calcium chloride, as shown in Figure 8a. The average inner and outer diameters of the resulting alginate microcapsules are 198 μm (CV = 4.6%) and 249 μm (CV = 3.3%). The citral with released from the alginate microcapsules continuously. The cumulative release curve is shown in Figure 8c. Four release models are used to explore the release behavior of microcapsules (Appendix A). The release behavior of citral from alginate microcapsules is more accordance with the first-order release model.

The slow-released property of citral-containing alginate microcapsules enables us to utilize citral in a sustained manner for different purposes, such as for antioxidants or air fresheners.

### 3.7. Improving of Antioxidant Properties of Oil by Citral

The changes in POVs of soybean oil with citral-containing alginate microcapsules, dissolved citral, and without citral are shown in Figure 9. All the POVs of soybean oil increase with a different rate because of the exposure to air. After one week, the POV of soybean oil with encapsulated citral, dissolved citral, and without citral increased gradually from 0.75 to 1.35, 0.75 to 1.42, and 0.75 to 2.04, respectively. The POV increasing rates of soybean oil with encapsulated citral or dissolved citral are nearly the same but evidently slower than the one without citral. The antioxidation property of citral in soybean oil slows the POV increasing rate. In 8–15 days, the POV of soybean oil with dissolved citral shows an accelerated rise, which is higher than the one with encapsulated citral. The encapsulation of citral in alginate shells will slow the volatilization and enables the sustained release of citral, which could prolong the antioxidation in soybean oil.

## 4. Conclusions

Citral is a typical active hydrophobic chemical and widely used in several areas. However, citral is sensitive to some environmental factors, such as acid, UV light, and high temperature, which will trigger deterioration by a series of cyclization and oxidation reactions. Thus, we proposed a mild method, without the using of UV irritation or acetic acid, to encapsulate citral in alginate microcapsules. Monodispersed O/W/O double emulsions are prepared with a microfluidic device as precursors, in which the middle aqueous phase contains sodium alginate, EDTA-Ca as calcium source, and GDL as acidifier. The delayed hydrolysis of GDL in aqueous solution will decrease the pH and trigger the calcium ions released from EDTA-Ca. By comprehensive consideration of the microfluidic operation and stable gelling of O/W/O emulsions into microcapsules, the concentration of the alginate sodium, EDTA-Ca, and GDL are optimized to 2% (*w*/*v*), 0.1 M and 2% (*w*/*v*), respectively. However, the pre-mixing of GDL with alginate-contained solution makes the microfluidic take the risk of blocking after a period of time of emulsion preparation. An on-line mixing of GDL with alginate-contained solution by a Y-shape micro-mixer as middle fluid for O/W/O emulsions preparation is also proposed, which may be preferred in continuous and large-scale industrial production. Citral is encapsulated into alginate microcapsules successfully by this method. The sustained-release characteristic of citral from alginate microcapsules is analyzed. Furthermore, a typical application of citral-containing alginate microcapsules to delay the oxidation of oil is also demonstrated. Compared with the traditional spray-drying method, microfluidic encapsulation technology will take a longer reaction time and shows limited production capacity. However, the volume of production could be raised via parallelized microfluidic platforms [48]. This mild gelation strategy, which is friendly to encapsulated substances, provides us a chance to encapsulate and utilize sensitive hydrophobic substances in alginate microcapsules for food, cosmetics, or pharmaceutical applications.

## Figures and Tables

**Figure 1 polymers-14-01165-f001:**
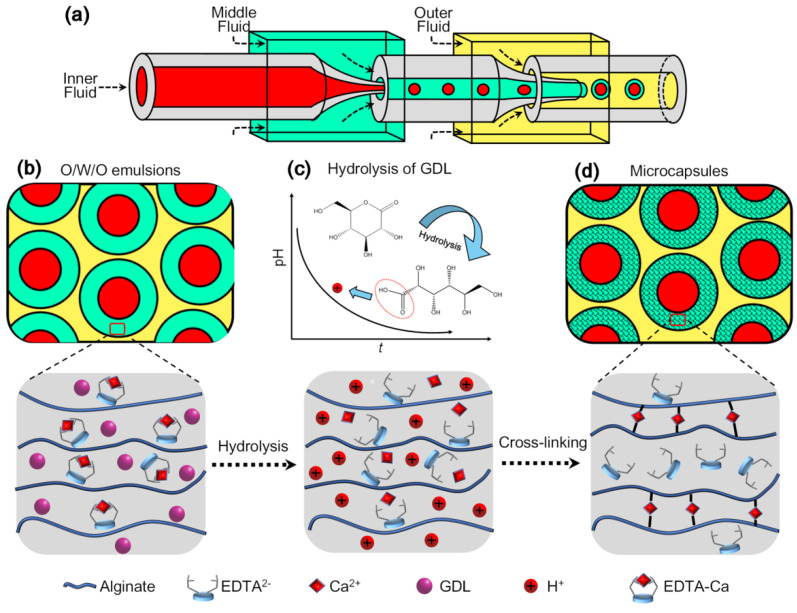
Schematic illustration for the preparation of alginate microcapsules under mild conditions. Monodispersed oil-in-water-in-oil (O/W/O) emulsions are generated in a two-staged microfluidic device (**a**). The middle aqueous phase of O/W/O emulsions contains sodium alginate, calcium–ethylenediaminetetraacetic acid (EDTA-Ca) as calcium source, and D-(+)-Gluconic acid δ-lactone (GDL) as acidifier (**b**). The hydrolysis of GDL in aqueous solution will decrease the pH value and trigger the calcium ions released from the EDTA-Ca complex (**c**). After the alginate molecules are cross-linked with the liberated calcium ions, the O/W/O emulsions will convert to alginate microcapsules (**d**). Citral could be encapsulated into alginate microcapsules by dissolving it in the inner fluid beforehand.

**Figure 2 polymers-14-01165-f002:**
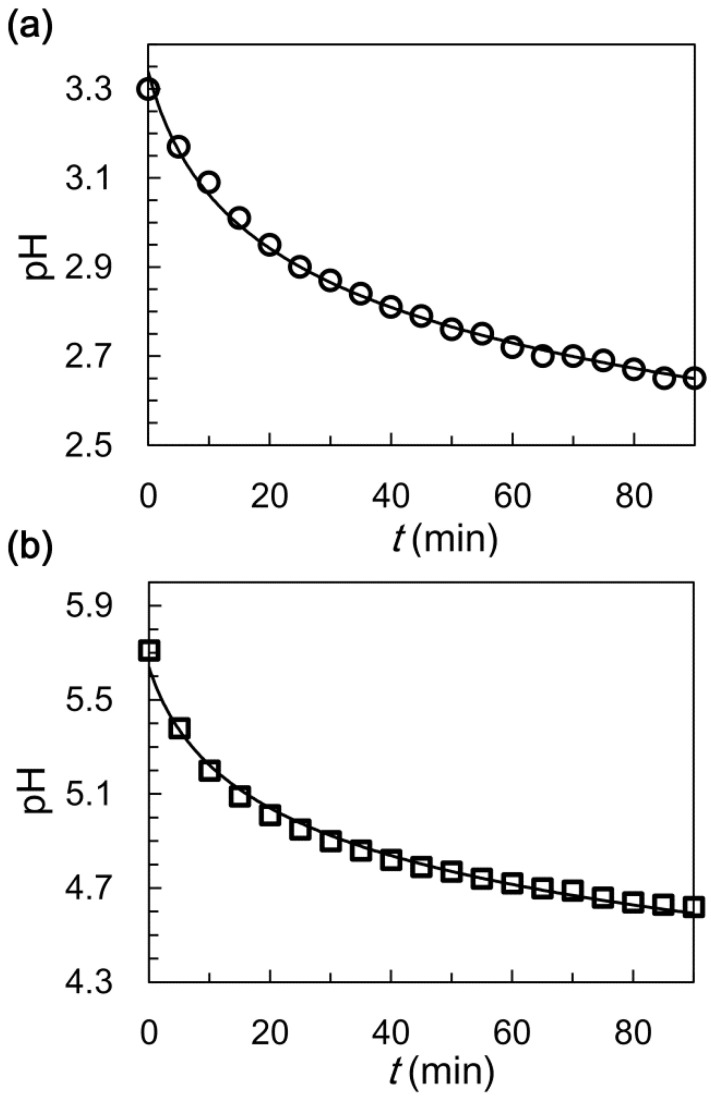
pH decrease induced by the hydrolysis of GDL. The pH variation process of aqueous solution with GDL (2%, *w*/*v*) (**a**) and the one with sodium alginate (2%, *w*/*v*), EDTA-Ca (0.1 M) and GDL (2%, *w*/*v*) (**b**).

**Figure 3 polymers-14-01165-f003:**
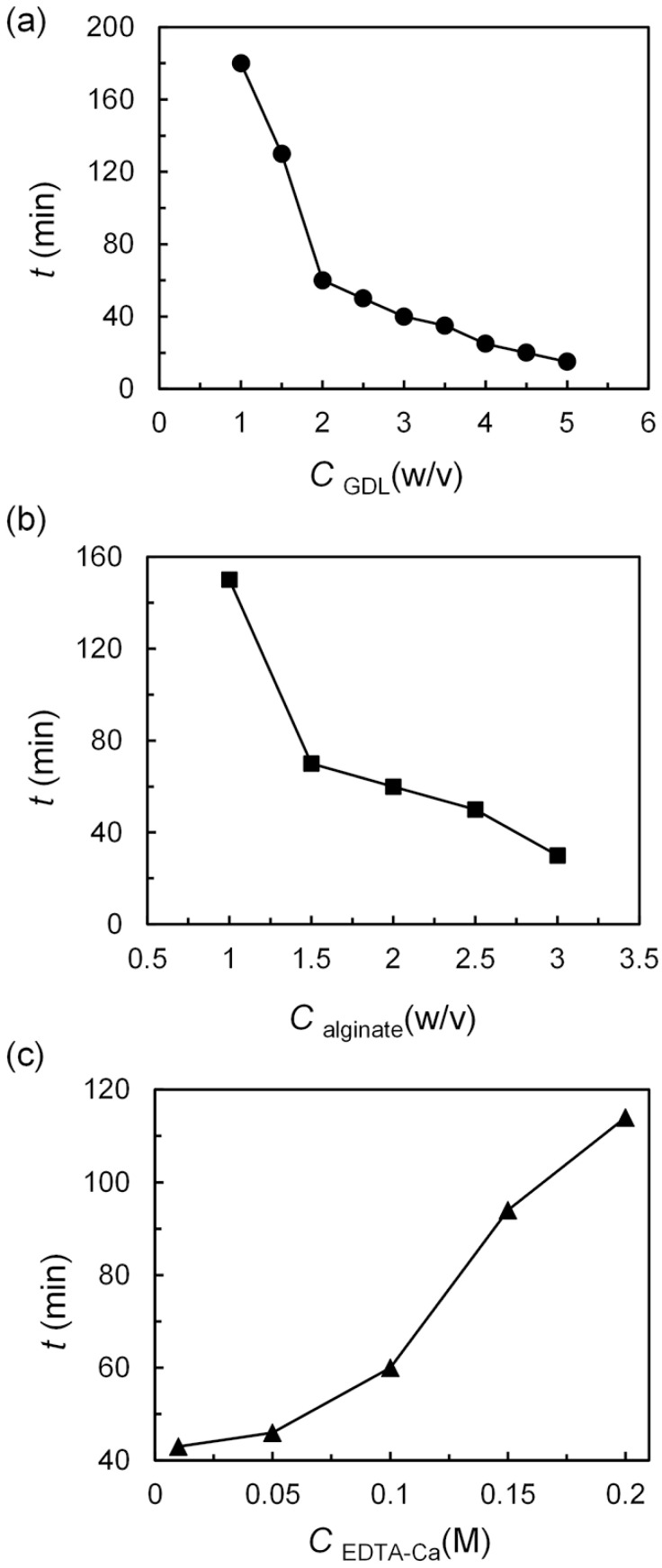
The influence of GDL (**a**), sodium alginate (**b**), and EDTA-Ca (**c**) on the gelling time.

**Figure 4 polymers-14-01165-f004:**
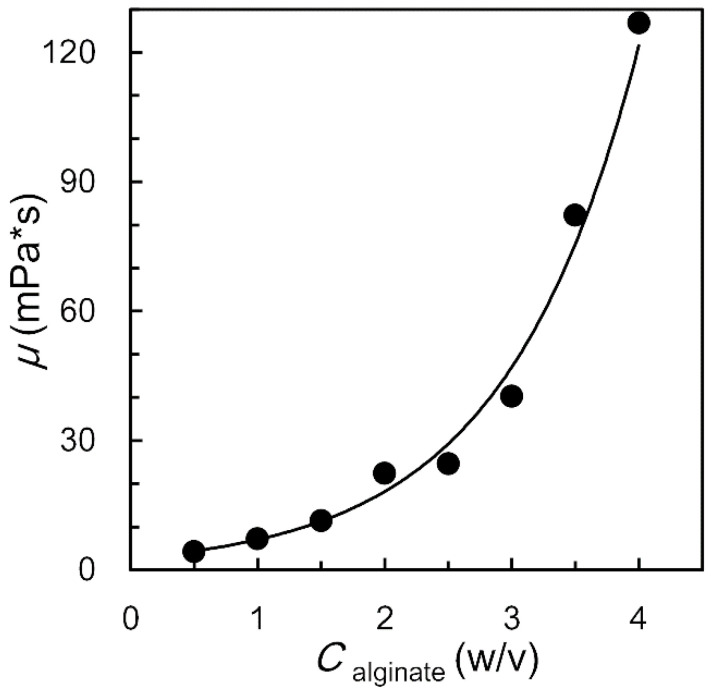
The viscosity of the aqueous solution with different sodium alginate content.

**Figure 5 polymers-14-01165-f005:**
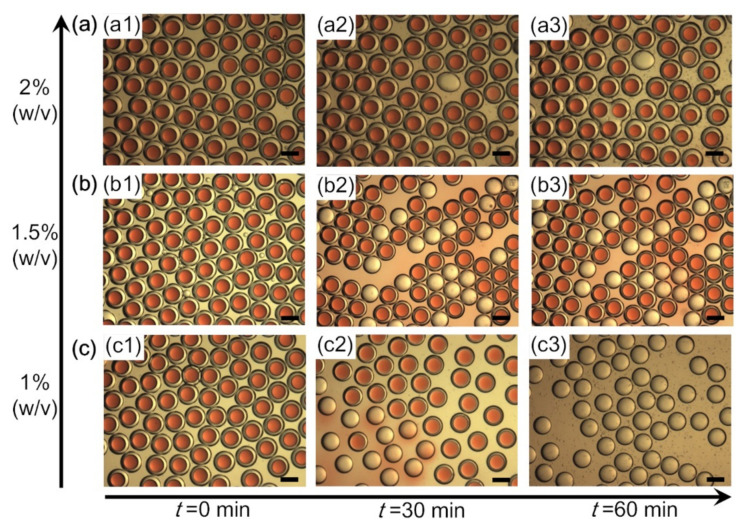
The influence of GDL content on the gelling process. The concentrations of alginate sodium and EDTA-Ca in the middle aqueous layer of O/W/O emulsions are fixed at 2% (*w*/*v*) and 0.1 M, respectively, while that of the GDL is varied from 2% (*w*/*v*) to 1% (*w*/*v*) (**a**–**c**). Scale bars are 200 μm.

**Figure 6 polymers-14-01165-f006:**
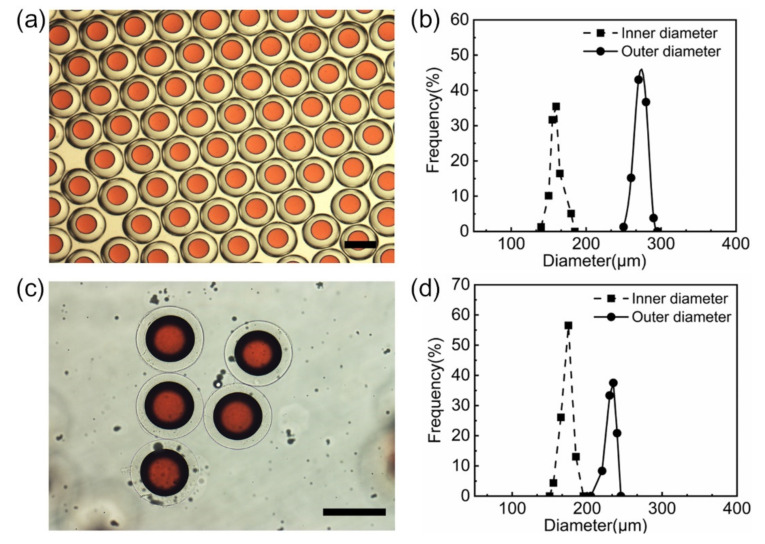
Preparation of O/W/O emulsions and alginate microcapsules by pre-mixing method. Optical micrographs (**a**) and the size distribution of O/W/O emulsions (**b**). The resultant alginate microcapsules 1% (*w*/*v*) CaCl_2_ solution (**c**) and the corresponding size distribution (**d**). Scale bars are 200 μm.

**Figure 7 polymers-14-01165-f007:**
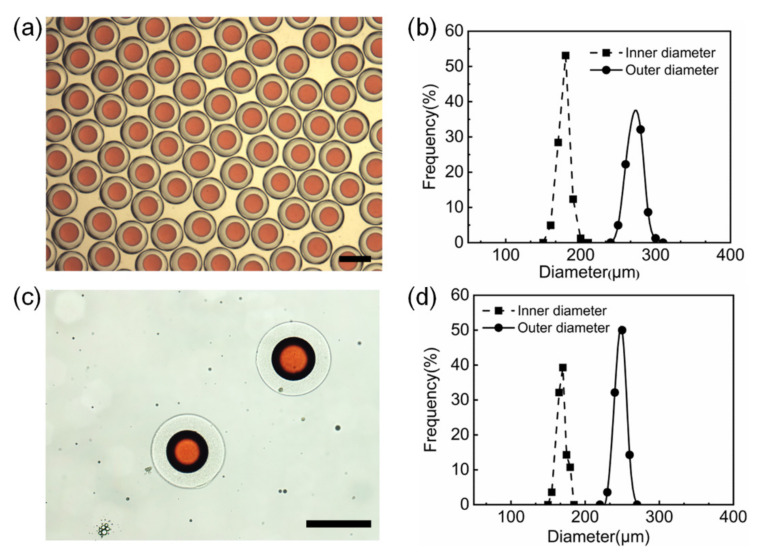
Preparation of microcapsules with micro-mixer by on-line mixing. Optical micrographs (**a**) and the size distribution of O/W/O emulsions (**b**). The resultant alginate microcapsules 1% (*w*/*v*) CaCl_2_ solution (**c**) and the corresponding size distribution (**d**). Scale bars are 200 μm.

**Figure 8 polymers-14-01165-f008:**
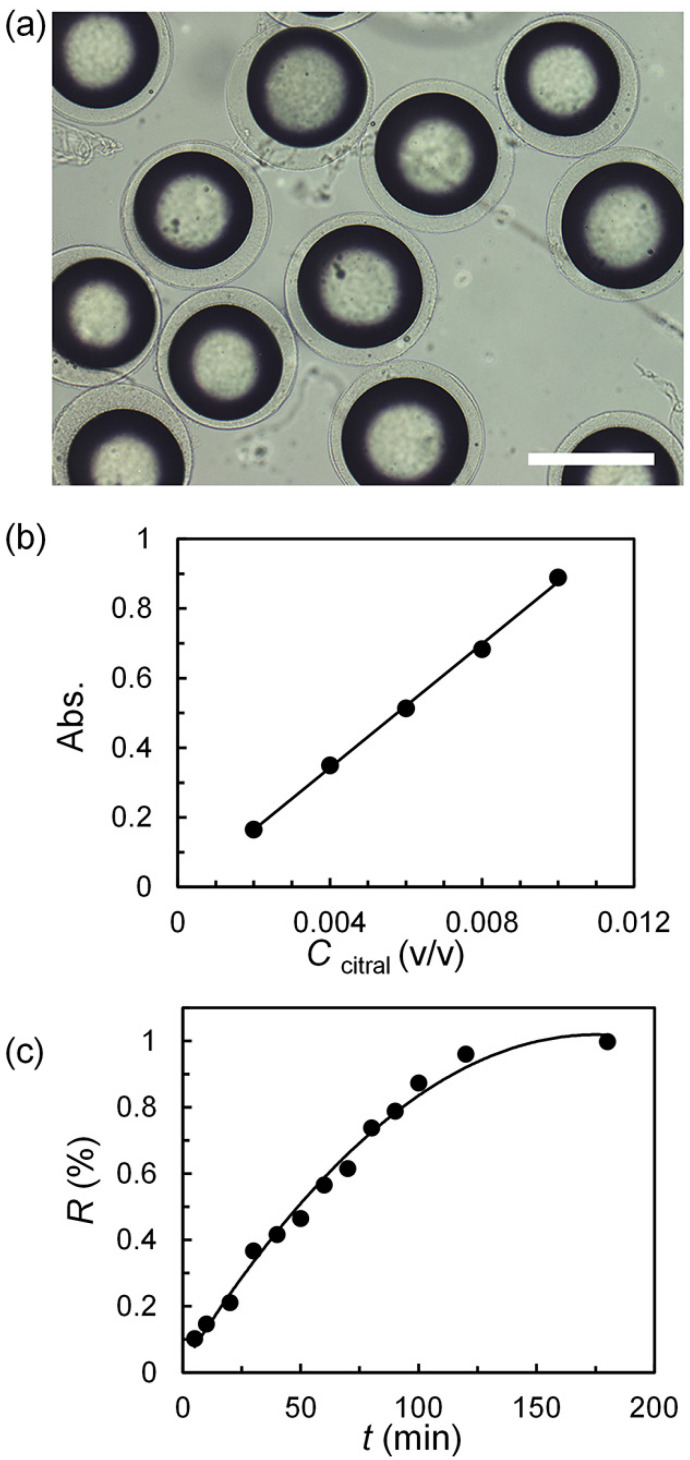
Sustained releasing properties of citral-containing alginate microcapsules. (**a**) Optical micrographs of citral-containing alginate microcapsules. The standard curve (**b**) and the release kinetics of citral from the alginate microcapsules (**c**). Scale bars are 200 μm.

**Figure 9 polymers-14-01165-f009:**
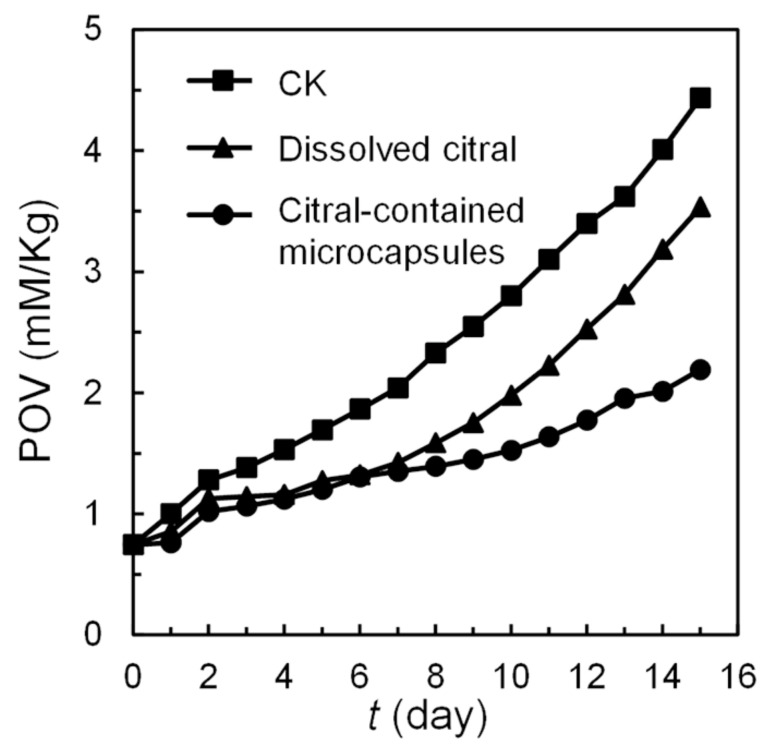
The peroxide values (POVs) of soybean oil with citral-containing alginate microcapsules, dissolved citral and without citral.

## Data Availability

Data available on request.

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
