# Peer review of "A Mild Method for Encapsulation of Citral in Monodispersed Alginate Microcapsules"

_polymers, 2022, doi:10.3390/polym14061165_

Round 1

Reviewer 1 Report

The paper describes the formation of a citral encapsulation in sodium alginate microcapsules. In general, the document is well written. However, it is desirable that authors improve some points before publication:

Authors should include the disadvantage of the microfluidic encapsulation technique vs spray drying. For example, production times and volume.

The authors describe the general applications of encapsulation systems; however, little information is described of the specific use of citral in O/W/O. What would be the specific application of this microencapsulated citral system? Why was the release of the active compound in ethanol solution? justify

Authors should describe the adverse effects of some of the substances used. For example, the concentration of EDTA-Na appears to be high for use. What is your recommended dosage for safe use?

Line 360-370 The authors mention that the internal drops coalescence to low with 1% of GDL. However, no significant change in droplet size is observed so coalescence may not be the dominant phenomenon. Authors should present evidence of the rate of coalescence of emulsion droplets.

Author Response

Response to Reviewer 1 Comments

Point 1: The paper describes the formation of a citral encapsulation in sodium alginate microcapsules.  In general, the document is well written.  However, it is desirable that authors improve some points before publication.

Response 1: The authors thank the Reviewer 1 very much for the positive comments.

Point 2: Authors should include the disadvantage of the microfluidic encapsulation technique vs spray drying.  For example, production times and volume.

Response 2: Many thanks for the valuable suggestion.  Compared with traditional spray drying method, microfluidic encapsulation technology will take longer reaction time and shows limited production capacity.  However, the volume of production could be raised via parallelized microfluidic platforms.( Refs: Jeong, H.H.; Issadore, D.; Lee, D. Recent developments in scale-up of microfluidic emulsion generation via parallelization. Korean J. Chem. Eng. 2016, 33, 1757-1766, doi:10.1007/s11814-016-0041-6).

In the Line 535-537, we have added the technical features of microfluidic in the revised manuscript. “Compared with traditional spray drying method, microfluidic encapsulation technology will take longer reaction time and shows limited production capacity. However, the volume of production could be raised via parallelized microfluidic platforms ”.

Point 3: The authors describe the general applications of encapsulation systems; however, little information is described of the specific use of citral in O/W/O.  What would be the specific application of this microencapsulated citral system?  Why was the release of the active compound in ethanol solution? Justify.

Response 3: Thank you for pointing this out to us.  

The citral contained microcapsules take many application in food spoilage inhibition ( Refs: Ju, J.; Xie, Y.F.; Yu, H.; Guo, Y.H.; Cheng, Y.L.; Qian, H.; Yao, W.R. A novel method to prolong bread shelf life: Sachets containing essential oils components. LWT-food science and technology 2020, 131, doi:10.1016/j.lwt.2020.109744, Miss-Zacarías, D.M.; Iñiguez-Moreno, M.; Calderón-Santoyo, M.; Ragazzo-Sánchez, J.A. Optimization of ultrasound-assisted microemulsions of citral using biopolymers: characterization and antifungal activity. Journal of Dispersion Science and Technology 2020, 1-10, doi:10.1080/01932691.2020.1857264.) and air purification (Refs: Wang, S.; Ding, H.H.; Zhao, Y.S.; Li, Y.G.; Wang, W. Fabrication of Protective Textile with N-doped TiO2 Embedded Citral Microcapsule Coating and Its Air Purification Properties. Fibers and polymers 2020, 21, 334-342, doi:10.1007/s12221-020-9352-7.) etc.  In the Line 36-38, We have introduced the application of the citral contained microcapsules and related references is added in the revised manuscript.  

Citral shows poor solubility in water, which makes the citral released from the microcapsules are very slow in aqueous solution.  At the same time, the released citral is easy to volatilize.  The accuracy of experimental data will be greatly affected by citral volatilization due to the long-time release process.  Citral is soluble in organic solvents, such as ethanol.  Therefore, the sustained-release of citral contained microcapsules is commonly carried out in ethanol solution. (Refs: Zhou, Y.; Yin, X.Q.; Chen, J.; Feng, D.C.; Zhu, L. Encapsulation efficiency and release of citral using methylcellulose as emulsifier and interior wall material in composite polysaccharide microcapsules. Advances in Polymer Technology 2018, 37, 3199-3209, doi:10.1002/adv.22089)  In the Line 236-241, we explained the reason for carrying out the sustained-release experiment in ethanol solution in the revised manuscript.

Point 4: Authors should describe the adverse effects of some of the substances used. For example, the concentration of EDTA-Na appears to be high for use.  What is your recommended dosage for safe use?  

Response 4: Thanks for your valuable counsel.  We have discussed the influence of EDTA-Na concentration on the formation of alginate microcapsules in Section 3.2 and 3.3.  The formed alginate microcapsules are separated from the collection solution by calcium chloride solution (0.1%, w/v) and DI water.  EDTA-Na is water soluble and will be removed in the repeated washing process.  In Section 3.2, the appropriate EDTA-Na concentration for alginate microcapsules preparation is 0.1 M.

In revised manuscript, we have further concretized and explained washing method in the Line 227-230.

Point 5: Line 360-370 The authors mention that the internal drops coalescence to low with 1% of GDL. However, no significant change in droplet size is observed so coalescence may not be the dominant phenomenon. Authors should present evidence of the rate of coalescence of emulsion droplets.  

Response 5: We feel great thanks for your professional review work on our article.

The coalescence of inner droplets with the continuous phase is a typical form of demulsification during emulsion polymerization process. (Refs:Mou, C.L.; Wang, W.; Li, Z.L.; Ju, X.J.; Xie, R.; Deng, N.N.; Wei, J.; Liu, Z.; Chu, L.Y. Trojan-Horse-Like Stimuli-Responsive Microcapsules. Advanced science 2018, 5, 1700960, doi:10.1002/advs.201700960.)  In the experiment, we have observed that the inner oil droplets get across the middle aqueous layer and merged into the continuous oil phase evry fast, less than 1 second.  In Figure 5c2, colored oil could be observed around the demulsified emulsions, which dues to coalescence of the red inner oil droplets with the continuous oil phase.  Furthermore, we have calculated the outer diameters of W/O emulsions based on the size of the O/W/O emulsions and the inner oil droplets by volume.  The outer diameter of O/W/O emulsions will decrease 11.9% after demulsificating to W/O emulsions.  Therefore, the size of the emulsions will will not vary distinctly by naked eye observation.  We also measured the outer diameters of O/W/O emulsions and the corresponding W/O emulsions.  The result shows that the average outer diameter of O/W/O emulsions and the corresponding W/O emulsions are 311.7 μm and 274.4 μm respectively.  The diameter of the W/O emulsions is 12.0% less than that of the O/W/O emulsions, which is basically consistent with the results theoretical calculation. 

In the Line 384-395, We have added the color and size change of the emulsions during the demulsification process in the Section 3.3.

We again thank the Reviewer 1 very much for the positive comments and the valuable suggestions.

Reviewer 2 Report

Dear Authors,

A few aspects of your experiment need to be improved.

  1. English language requires correction.
  2. Materials and Methods – Sections 2.3 and 2.4. Part concerning the preparation of alginate aqueous solution was repeated, is it necessary?
  3. Materials and Methods – Section 2.4. “The concentration of sodium alginate in mixture is varied from 2.0% (w/v) to 1.0% (w/v), 1.5% (w/v), 2.5% (w/v) and 3.0% (w/v) and the gelling time …” There is a problem with phrase “varied from” – you wrote a range and then you wrote another concentration, which are beyond the range. This problem was observed a few lines below.
  4. Results – Sections 3.3. Description of Figure 5 – 1% of GDL is shown in Figure 5 c (in text there is information Fig. 5 a).

Author Response

Response to Reviewer 2 Comments

Point 1: English language requires correction.

Response 1: Thanks for the valuable suggestion.  To address the comment, relevant corrections and revisions have been made in the revised version, and we have also gotten some help from a native English speaker to correct and double-check the grammar. 

Point 2: Materials and Methods – Sections 2.3 and 2.4. Part concerning the preparation of alginate aqueous solution was repeated, is it necessary?

Response 2:  Many thanks for the valuable suggestion.  In the Sections 2.4, the description of solution preparation is simplified. ”Aqueous solution with alginate(2%, w/v), EDTA-Ca (0.1 M) and GDL 2% (w/v) is prepared with the same method as Section 2.3.”

Point 3: Materials and Methods – Section 2.4. “The concentration of sodium alginate in mixture is varied from 2.0% (w/v) to 1.0% (w/v), 1.5% (w/v), 2.5% (w/v) and 3.0% (w/v) and the gelling time …” There is a problem with phrase “varied from” – you wrote a range and then you wrote another concentration, which are beyond the range. This problem was observed a few lines below

Response 3: Thanks for your valuable counsel.  The description of the concentration range has been corrected in the revised manuscript. 

In the Line 168-172, “The concentration of GDL in the mixture is adjusted from 0.5% (w/v) to 5.0% (w/v), while the concentration of other components remains unchanged.”

In the Line 175-178, “The concentration of sodium alginate in mixture is varied from 1.0% (w/v) to 3.0% (w/v) and the gelling time is compared.”

In the Line 180-183, “The concentrations of calcium chloride and disodium–EDTA are varied from 0.01 M to 0.20 M and the gelling time is also compared.”

In the Line 197-202, ”The concentration of GDL in the middle fluid is varied from 1.0% (w/v) to 3.0% (w/v), or that of the sodium alginate is varied from 1.0% (w/v) to 3.0% (w/v).”

Point 4: Results – Sections 3.3. Description of Figure 5 – 1% of GDL is shown in Figure 5 c (in text there is information Fig. 5 a).

Response 4: Thank you for pointing this out.  

The title of Figure 5 has been corrected.  “The influence of GDL content on the gelling process.  The concentrations of alginate sodium and EDTA-Ca in the middle aqueous layer of O/W/O emulsions are fixed at 2.0% (w/v) and 0.10 M respectively, while that of the GDL is varied from 2% (w/v) to 1% (w/v) (a-c).  Scale bars are 200 μm.”

In the Line 384, 397, the description of Figure 5 in the main text has also been corrected in the revised manuscript.

We again thank the Reviewer 2 very much for the positive comments and the valuable comments and suggestions.
